# Approximate Representations of Shaped Pulses Using the Homotopy Analysis Method

Timothy Crawley[1] and Arthur G. Palmer, III[1]

[1]Department of Biochemistry and Molecular Biophysics, Columbia University, 630 West 168th Street, New York, NY 10032, United States

**Correspondence:** Arthur G. Palmer, III (agp6@columbia.edu)

**Abstract.** The evolution of nuclear spin magnetization during a radiofrequency pulse in the absence of relaxation or coupling interactions can be described by three Euler angles. The Euler angles in turn can be obtained from the solution of a Riccati differential equation; however, analytic solutions exist only for rectangular and chirp pulses. The Homotopy Analysis Method is used to obtain new approximate solutions to the Riccati equation for shaped radiofrequency pulses in NMR spectroscopy. The results of even relatively low orders of approximation are highly accurate and can be calculated very efficiently. The results are extended in a second application of the Homotopy Analysis Method to represent relaxation as a perturbation of the magnetization trajectory calculated in the absence of relaxation. The Homotopy Analysis Method is powerful and flexible and is likely to have other applications in magnetic resonance.

## 1 Introduction

Numerous aspects of NMR spectroscopy are formulated in terms of differential equations, few of which have closed-form analytical solutions. In an era characterized by ever-increasing compuntional capabilities, numerical solutions to such differential equations are always possible and frequently are the preferred approach for applications, such as data analysis. However, approximate solutions can provide useful formulas as well as insights difficult to discern from purely numerical results.

As one example, the net evolution of magnetization of an isolated spin during a radiofrequency pulse, i.e. in the absence of relaxation and scalar or other coupling interactions, can be described by three rotations with Euler angles $\alpha(\tau_p)$, $\beta(\tau_p)$, $\gamma(\tau_p)$, in which $\tau_p$ is the pulse length (Zhou et al., 1994; Siminovitch, 1997a, b). Shaped pulses, in which the amplitude (Rabi frequency), phase, or radiofrequency are time-dependent, are widely applied in modern NMR spectroscopy and other magnetic resonance techniques (Geen and Freeman, 1991; Emsley and Bodenhausen, 1992; Kupče et al., 1995; Cavanagh et al., 2007). The Euler angles for an arbitrary shaped pulse can be extracted from a numerical calculation in which the shaped pulse is represented by a series of $K$ short rectangular pulses with appropriate amplitudes and phases. Thus, the propagator for a

shaped pulse expressed in the Cartesian basis is given by (Siminovitch, 1995):

$$\mathbf{U} = e^{-i\gamma(\tau_p)I_z} e^{-i\beta(\tau_p)I_x} e^{-i\alpha(\tau_p)I_z} \tag{1}$$

$$= \begin{bmatrix} e^{-i(\alpha(\tau_p)+\gamma(\tau_p))/2}\cos(\beta(\tau_p)/2) & -ie^{i(\alpha(\tau_p)-\gamma(\tau_p))/2}\sin(\beta(\tau_p)/2) \\ -ie^{-i(\alpha(\tau_p)-\gamma(\tau_p))/2}\sin(\beta(\tau_p)/2) & e^{i(\alpha(\tau_p)+\gamma(\tau_p))/2}\cos(\beta(\tau_p)/2) \end{bmatrix} \tag{2}$$

$$= \prod_{k=1}^{K} \mathbf{U}_k \tag{3}$$

in which $I_k$ are the Cartesian spin operators, the product is time-ordered from right to left, and the propagator for the $k$th rectangular pulse segment is:

$$\mathbf{U}_k = \begin{bmatrix} \cos(\omega_e\Delta t_k/2) - i\cos\theta\sin(\omega_e\Delta t_k/2) & -ie^{-i\phi}\sin(\omega_e\Delta t_k/2) \\ -ie^{i\phi}\sin(\omega_e\Delta t_k/2) & \cos(\omega_e\Delta t_k/2) + i\cos\theta\sin(\omega_e\Delta t_k/2) \end{bmatrix} \tag{4}$$

In this expression, $\omega_{1k}$, $\phi_k$ and $\Delta t_k$ are the radiofrequency field strength, phase angle, and duration of the $k$th pulse segment; $\Omega_k$ is the resonance offset in the rotating frame of reference during the $k$th pulse segment (and is constant if the offset is fixed);

$\omega_e = (\omega_{1k}^2 + \Omega_k^2)^{1/2}$ is the effective field; and $\theta = \tan^{-1}(\omega_{1k}/\Omega_k)$ is the tilt angle. Values of $\alpha(\tau_p)$, $\beta(\tau_p)$, and $\gamma(\tau_p)$ then are obtained from the matrix elements of $\mathbf{U}$.

Alternatively, the Euler angles can be determined from the solution of a Ricatti equation (Zhou et al., 1994):

$$\frac{df(t)}{dt} = \frac{1}{2}\omega^+(t)f^2(t) + i\Omega(t)f(t) + \frac{1}{2}\omega^-(t) \tag{5}$$

in which:

$$f(t) = \tan\left(\frac{\beta(t)}{2}\right)e^{i\gamma(t)} \tag{6}$$

$\omega^\pm(t) = \omega_x(t) \pm i\omega_y(t)$ and $\omega_x(t)$ and $\omega_y(t)$ are the Cartesian amplitude components of the radiofrequency field in the rotating frame of reference. After solution of the Riccati equation, $\beta(\tau_p)$ and $\gamma(\tau_p)$ are obtained from the magnitude and argument of $f(\tau_p)$ and the value of $\alpha(\tau_p)$ is obtained by integration:

$$\alpha(\tau_p) = \int_0^{\tau_p} dt\{\omega_x(t)\sin[\gamma(t)] - \omega_y(t)\cos[\gamma(t)]\}/\sin[\beta(t)] \tag{7}$$

The Riccati equation can be transformed into a second-order differential equation:

$$\frac{d^2y(t)}{dt^2} - \left[\frac{d\ln[\omega^-(t)]}{dt} + i\Omega(t)\right]\frac{dy(t)}{dt} + \frac{1}{4}|\omega(t)|^2y(t) = 0 \tag{8}$$

by use of the definition:

$$\frac{d\ln[y(t)]}{dt} = -\frac{1}{2}\omega^-(t)f(t) \tag{9}$$

A more compact form is obtained by defining:

$$\hat{\omega}^-(t) = exp\left[i\int_0^t \Omega(t')dt'\right]\omega^-(t) \tag{10}$$

to yield:

$$\frac{d^2y(t)}{dt^2} - \frac{dln\left[\hat{\omega}^-(t)\right]}{dt}\frac{dy(t)}{dt} + \frac{1}{4}|\hat{\omega}(t)|^2y(t) = 0 \tag{11}$$

The Riccati differential equation only can be solved analytically for a single rectangular or chirp pulse. Approximate solutions for arbitrary shaped pulses have been derived by perturbation theory for the limits of small, using Eq. (11), and large, using Eq. (5), resonance offsets (Li et al., 2014); however, perturbation theory is unwieldly to apply to high order, and obviously depends on the perturbation parameters being small, in some respect.

The Homotopy Analysis Method (HAM) is a fairly recent development, first reported in 1992 (Liao, 1992), for approximating solutions to differential equations, particularly non-linear ones. HAM does not depend on small parameters, unlike perturbation theory, and has proven powerful in a number of applications outside of NMR spectroscopy (Liao, 2012). The present paper illustrates HAM by application to the solutions of Eqs. (5) and (11) and subsequently in an extension to the Bloch equations, including relaxation.

## 2  Theory

In topology, a pair of functions defining different topological spaces are said to be homotopic if the shape defined by one function can be continuously transformed (deformed in the lexicon of topology) into the shape defined by the other. Analogously, the essence of HAM is to map a function of interest, here $y(t)$ (or $f(t)$), to a second function, $\Phi(t;q)$, which has a known solution and is a function of both $t$ and an embedding parameter $q \in [0,1]$.

This relationship is established by constructing the homotopy (Liao, 2012):

$$\mathcal{H}\left[\Phi(t;q):q\right] = (1-q)\mathcal{L}\left[\Phi(t;q)-y_0(t)\right] - qc_0H(t)\mathcal{N}\left[\Phi(t;q)\right] \tag{12}$$

in which $\mathcal{L}[\,]$ is a linear (differential) operator and $\mathcal{N}[\,]$ is an (non-linear differential) operator satisfying,

$$\mathcal{L}[0] = 0 \tag{13}$$

$$\mathcal{N}[y(t)] = 0 \tag{14}$$

$y_0(t)$ is an initial approximation for the desired solution $y(t)$, $c_0 \neq 0$ is a convergence control parameter and $H(t) \neq 0$ is an auxiliary function (vide infra). From the homotopy equation, when $q = 0$,

$$\mathcal{H}\left[\Phi(t;0):0\right] = \mathcal{L}\left[\Phi(t;0)-y_0(t)\right] \tag{15}$$

Therefore, when $\mathcal{H}\left[\Phi(t;0):0\right] = 0$, Eq. (13) requires $\Phi(t,0) = y_0(t)$. Similarly, when $q = 1$,

$$\mathcal{H}\left[\Phi(t;1):1\right] = -c_0H(t)\mathcal{N}\left[\Phi(t,1)\right] \tag{16}$$

Therefore, when $\mathcal{H}\left[\Phi(t;1):1\right] = 0$, Eq. (14) requires $\Phi(t,1) = y(t)$. Stated more succinctly, as $q$ increases from $0 \to 1$, $\Phi(t;q)$ deforms from the initial approximation $y_0(t)$ to the exact solution $y(t)$. To proceed, the Maclaurin series for $\Phi(t;q)$ is assumed to exist; conditions concerning convergence of the series are discussed by Liao (Liao, 2012):

$$\Phi(t;q) = \sum_{n=0}^{\infty} y_n(t)q^n \tag{17}$$

in which

$$y_n(t) = \frac{1}{n!}\frac{d^n\Phi(t;q)}{dq^n}\bigg|_{q=0} \tag{18}$$

Equation (17) has the desired properties $\Phi(t;0) = y_0(t)$ and

$$\Phi(t;1) = y(t) = \sum_{n=0}^{\infty} y_n(t) \tag{19}$$

HAM then consists of successively determining $y_n(t)$, beginning with the initial approximation $y_0(t)$, until $y(t)$ is approximated to desired accuracy. The choices of $\mathcal{L}[\,]$, $y_0(t)$, $c_0$, and $H(t)$ provide considerable flexibility in finding approximate solutions to differential equations. For simplicity in the following, the auxiliary function $H(t) = 1$.

The iterative algorithm in HAM is illustrated by application to the second-order differential form of the Riccati equation. In the first example, the non-linear operator is obtained from Eq. (11):

$$\mathcal{N}\left[g(t)\right] = \frac{d^2g(t)}{dt^2} - \frac{dln\left[\hat{\omega}^-(t)\right]}{dt}\frac{dg(t)}{dt} + \frac{1}{4}|\hat{\omega}(t)|^2 \tag{20}$$

in which $g(t)$ is an arbitrary function. The linear operator is chosen to be:

$$\mathcal{L}\left[g(t)\right] = \frac{d^2g(t)}{dt^2} - \frac{dln\left[\hat{\omega}^-(t)\right]}{dt}\frac{dg(t)}{dt} \tag{21}$$

and the initial approximation is $y_0(t) = 1$.

From the relationships of Eqs. (13) and (14) embedded in the initial homotopy, Eq. (12), the zeroth-order deformation
equation is defined as (Liao, 2012):

$$(1-q)\mathcal{L}\left[\Phi(t;q) - y_0(t)\right] = qc_0\mathcal{N}\left[\Phi(t;q)\right] \tag{22}$$

The derivative of Eq. (22) with respect to $q$ yields the first-order deformation equation:

$$-\mathcal{L}\left[\Phi(t;q) - y_0(t)\right] + (1-q)\mathcal{L}\left[\frac{d\Phi(t;q)}{dq}\right] = c_0\mathcal{N}\left[\Phi(t;q)\right] + qc_0\frac{d}{dq}\mathcal{N}\left[\Phi(t;q)\right] \tag{23}$$

The limit $q \to 0$ gives:

$$-\mathcal{L}\left[\Phi(t,0) - y_0(t)\right] + \mathcal{L}\left[\frac{d\Phi(t;q)}{dq}\bigg|_{q=0}\right] = c_0\mathcal{N}\left[\Phi(t,0)\right]$$

$$\mathcal{L}\left[y_1(t)\right] = c_0\mathcal{N}\left[y_0(t)\right] \tag{24}$$

in which the second line is obtained using $\Phi(t,0) = y_0(t)$ and Eq. (18). Substituting for $\mathcal{N}[]$, $\mathcal{L}[]$, and $y_0(t)$ yields

$$\frac{d^2y_1(t)}{dt^2} - \frac{dln\,[\hat{\omega}^-(t)]}{dt}\frac{dy_1(t)}{dt} = c_0\left(\frac{d^2y_0(t)}{dt^2} - \frac{dln\,[\hat{\omega}^-(t)]}{dt}\frac{dy_0(t)}{dt} + \frac{1}{4}|\hat{\omega}(t)|^2y_0(t)\right)$$

$$= \frac{c_0}{4}|\hat{\omega}(t)|^2 \tag{25}$$

in which the second line is obtained using $dy_0(t)/dt = 0$. This differential equation does not contain a term proportional to $y_1(t)$. Hence, the homogenous equation (setting the right-hand side to 0) can be solved by two successive integrations and the inhomogeneous solution obtained by the technique of variation of parameters (Arfken et al., 2013). The solution is:

$$y_1(t) = \frac{c_0}{4}\int_0^t \hat{\omega}^-(t')\int_0^{t'} \hat{\omega}^+(t'')dt''dt' \tag{26}$$

The higher-order approximations $y_n(t)$ are obtained in similar fashion. The $n$th derivative with respect to $q$ of Eq. (22) yields (for $n > 1$):

$$-n\mathcal{L}\left[\frac{d^{n-1}\Phi(t;q)}{dq^{n-1}}\right] + (1-q)\mathcal{L}\left[\frac{d^n\Phi(t;q)}{dq^n}\right] = nc_0\frac{d^{n-1}}{dq^{n-1}}\mathcal{N}[\Phi(t;q)] + qc_0\frac{d^n}{dq^n}\mathcal{N}[\Phi(t;q)] \tag{27}$$

Executing the derivatives, taking the limit $q \to 0$, and dividing both sides of the equation by $n!$ gives:

$$\frac{d^2y_n(t)}{dt^2} - \frac{dln\,[\hat{\omega}^-(t)]}{dt}\frac{dy_n(t)}{dt} = (c_0+1)\{\frac{d^2y_{n-1}(t)}{dt^2} - \frac{dln\,[\hat{\omega}^-(t)]}{dt}\frac{dy_{n-1}(t)}{dt}\} + \frac{1}{4}c_0|\hat{\omega}(t)|^2y_{n-1}(t) \tag{28}$$

with the solution obtained by the same approach as for Eq. (26):

$$y_n(t) = (c_0+1)y_{n-1}(t) + \frac{c_0}{4}\int_0^t \hat{\omega}^-(t')\int_0^{t'} \hat{\omega}^+(t'')y_{n-1}(t'')dt''dt' \tag{29}$$

Successive use of Eqs. (26) and (29) allows $y(t)$ and hence $f(t)$ to be determined to arbitrary accuracy:

$$f(t) = \left(\frac{-2}{\omega^-(t)}\right)\frac{dln\,[y(t)]}{dt} = \left(\frac{-2}{\omega^-(t)}\right)\frac{\sum_{m=0}^N \frac{dy_m(t)}{dt}}{\sum_{n=0}^N y_n(t)} \tag{30}$$

in which $N$ is the order of approximation. For completeness, the derivatives of Eqs. (26), and (29) are, respectively:

$$\frac{dy_1(t)}{dt} = \frac{c_0}{4}\hat{\omega}^-(t)\int_0^t \hat{\omega}^+(t')dt' \tag{31}$$

$$\frac{dy_n(t)}{dt} = (c_0+1)\frac{dy_{n-1}(t)}{dt} + \frac{c_0}{4}\hat{\omega}^-(t)\int_0^t \hat{\omega}^+(t')y_{n-1}(t')dt' \tag{32}$$

Results obtained using $y_0(t) = 1$ together with Eqs. (26) and (29-30) will be called Method 1 in the following discussion. The iterated form of the above expressions for $y_n(t)$ have similarities to the Fourier intergrals obtained from average Hamiltonian theory by Warren (Warren, 1984).

The above choices of $\mathcal{L}[]$ and $y_0(t)$ are not unique. Different choices lead to different series approximations and hence to different qualitative and quantitative results. As a second example, $\Omega(t) = \Omega$ is assumed to be fixed and only amplitude-modulated pulses $\omega(t)$ with $x$-phase are considered (these assumptions can be relaxed as needed). Returning to Eq. (8):

$$\mathcal{N}[g(t)] = \frac{d^2 g(t)}{dt^2} - \left[\frac{dln\,[\omega(t)]}{dt} + i\Omega\right]\frac{dg(t)}{dt} + \frac{1}{4}\omega^2(t)g(t) \tag{33}$$

$$\mathcal{L}[g(t)] = \frac{d^2 g(t)}{dt^2} - \frac{dln\,[\omega(t)]}{dt}\frac{dg(t)}{dt} + \frac{1}{4}\omega^2(t)g(t) \tag{34}$$

$$y_0(t) = \cos\left[\frac{1}{2}\delta(t)\right] \tag{35}$$

in which:

$$\delta(t) = \int\limits_0^t \omega(t')dt' \tag{36}$$

This choice of $y_0(t)$ satisfies:

$$\frac{d^2 y_0(t)}{dt^2} - \frac{dln\,[\omega(t)]}{dt}\frac{dy_0(t)}{dt} + \frac{1}{4}\omega^2(t)y_0(t) = 0 \tag{37}$$

and is the exact on-resonance solution for $y(t)$. Consequently, the first-order deformation equation leads to:

$$\frac{d^2 y_1(t)}{dt^2} - \frac{dln\,[\omega(t)]}{dt}\frac{dy_1(t)}{dt} + \frac{1}{4}\omega^2(t)y_1(t) = -ic_0\Omega\frac{dy_0(t)}{dt} \tag{38}$$

The solutions to the homogeneous equation (setting the right-hand-side to 0) are $y^\pm(t) = e^{\pm i\delta(t)/2}$. The method of variation of parameters then gives the inhomogeneous solution as:

$$y_1(t) = -ic_0\Omega\int\limits_0^t \sin\left[\frac{\delta(t)}{2} - \frac{\delta(t')}{2}\right]\sin\left[\frac{\delta(t')}{2}\right]dt' \tag{39}$$

The $n$th-order deformation equation for $n > 1$ is:

$$\frac{d^2 y_n(t)}{dt^2} - \left[\frac{dln\,[\omega(t)]}{dt} + i\Omega\right]\frac{dy_n(t)}{dt} + \frac{1}{4}\omega^2(t)y_n(t) =$$
$$(1+c_0)\{\frac{d^2 y_{n-1}(t)}{dt^2} - \frac{dln\,[\omega(t)]}{dt}\frac{dy_{n-1}(t)}{dt} + \frac{1}{4}\omega^2(t)y_{n-1}(t)\} - ic_0\Omega\frac{dy_{n-1}(t)}{dt} \tag{40}$$

with the solution:

$$y_n(t) = (1+c_0)y_{n-1}(t) - ic_0\Omega\int\limits_0^t \frac{2}{\omega(t')}\sin\left[\frac{\delta(t)}{2} - \frac{\delta(t')}{2}\right]\frac{dy_{n-1}(t')}{dt'}dt' \tag{41}$$

Each $y_n(t)$ is proportional to $\Omega^n$ and these results yield a power series in $\Omega$ for $y(t)$:

$$y(t) = y_0(t) + \sum_{n=1}^N (2+c_0)y_n(t) \tag{42}$$

which is substituted into Eq. (30) to obtain $f(t)$. Results using Eqs. (39), (41) and (42) will be called Method 2 in the following discussion. For completeness, the derivatives of Eqs. (39) and (41) are:

$$\frac{dy_1(t)}{dt} = -ic_0\Omega\frac{\omega(t)}{2}\int_0^t \cos\left[\frac{\delta(t)}{2} - \frac{\delta(t')}{2}\right]\sin\left[\frac{\delta(t')}{2}\right]dt' \tag{43}$$

$$\frac{dy_n(t)}{dt} = (1+c_0)\frac{y_{n-1}(t)}{dt} - ic_0\Omega\frac{\omega(t)}{2}\int_0^t \frac{2}{\omega(t')}\cos\left[\frac{\delta(t)}{2} - \frac{\delta(t')}{2}\right]\frac{dy_{n-1}(t')}{dt'}dt' \tag{44}$$

## 2.1 Methods

Numerical integration was performed using the trapezoid method, implemented in Python 3.6. Pulse shapes were discretized in 1000 increments. Rectangular pulses were simulated using $\omega_1/(2\pi) = 25,000$ Hz and an on-resonance 90° pulse length of 10.0 $\mu$s or $\omega_1/(2\pi) = 250$ Hz and an on-resonance 90° pulse length of 1 ms. Eburp-2 (Geen and Freeman, 1991) and Q5 (Emsley and Bodenhausen, 1992) pulses were simulated using a maximum $\omega_1/(2\pi) = 9,000$ Hz and 90° pulse lengths of 455.2 $\mu$s and 504.9 $\mu$s, respectively. REBURP (Geen and Freeman, 1991) pulses were simulated using a maximum $\omega_1/(2\pi) = 10,000$ Hz and a 180° pulse length of 626.5 $\mu$s. WURST-20 (Kupče and Freeman, 1995) pulses were were simulated using maximum $\omega_1/(2\pi) = 9512$ Hz, frequency sweep of 50,000 Hz, and a pulse length of 440.0 $\mu$s.

Equation (7) can be recast as:

$$\alpha(\tau_p) = \frac{i}{4}\int_0^{\tau_p} dt\{\omega^+(t)f^*(t) - \omega^-(t)f(t)\}\{1 + |f(t)|^2\}/|f(t)|^2 \tag{45}$$

for numerical calculations; $\alpha(\tau_p)$ also can be obtained from the argument of $f(\tau_p)$ calculated for the time-reversed pulse (Li et al., 2014). The latter is more computationally demanding, but more numerically stable, and was used for the results presented herein.

## 2.2 Results and Discussion

In the present applications, HAM converts the second-order Riccati differential equation, Eq. (8), which cannot be solved directly, into a series of second-order differential equations that have convenient solutions. The choice of $y_0(t) = 1$ leads using Method 1 to simple iterative solutions that can be calculated very efficiently. The form of $y_0(t)$ given in Eq. (35) also could be used in Eq. (24) to obtain an alternative expression for $y_1(t)$ to be substituted into Eqs. (29), and (30). The resulting first-order expressions for $y(t)$ are usually more accurate than the first-order results obtained using $y_0(t) = 1$, but this advantage becomes less pronounced at higher orders of approximation and comes at increased computational cost. Thus, Eqs. (26), (29), and (30) are most suitable in practice.

A first example of the results of the above analysis are given for a rectangular 90° pulse in Fig. 1. The integrals in Eqs. (26) and (29) can be performed analytically for a rectangular pulse with amplitude $\omega_1$. For example, using Eq. (26):

$$y_1(t) = \frac{c_0\omega_1^2}{4\Omega^2}\left(1 - e^{i\Omega t}\right) + i\frac{c_0\omega_1^2 t}{4\Omega} \tag{46}$$

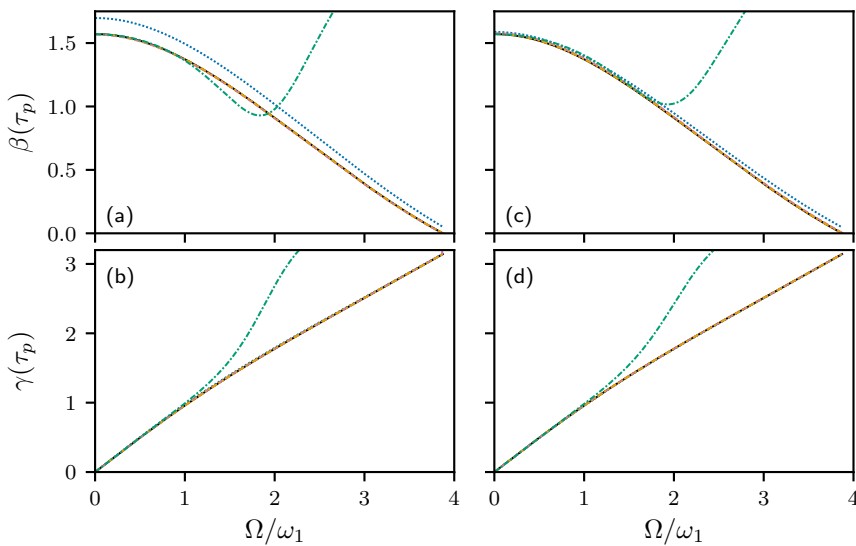

**Figure 1.** HAM approximations for on-resonance 90° rectangular pulse with $\omega_1/(2\pi) = 25,000$ Hz. (black) Exact calculation of Euler angles $\beta(\tau_p)$ and $\gamma(\tau_p)$. For a rectangular pulse, $\alpha(\tau_p) = \gamma(\tau_p)$. (blue, dotted) First-order, (reddish-purple, dashed) second-order, and (orange, dash-dot-dotted) third-order HAM results using Method 1. (green, dash-dotted) Third-order result using the power series of Method 2. Results are shown for (a,b) $c_0 = -1$ and (c,d) $c_0 = -0.925$. The exact, second-order HAM and third-order HAM curves for Method 1 are virtually indistinguishable.

however, analytic calculations of higher order $y_n(t)$ do not have advantages over numerical integration. As shown in Fig. 1a,b, the second- and third-order results obtained with Method 1 and $c_0 = -1$ are nearly indistinguishable from the exact result of Eq. (4) (using $\tau_p = \Delta\tau_k$) over the range of resonance offsets from 0 to $\Omega/\omega_1 = 15^{1/2}$. The first-order result provides a highly accurate estimate of $\gamma(\tau_p)$, but overestimates $\beta(\tau_p)$. The role of the convergence control parameter $c_0$ is illustrated in Fig. 1c,d. A value of $c_0 = -0.925$ was chosen using Eqs. (46) and (30) to scale the first-order result for $\beta(\tau_p)$ to be equal to $\pi/2$ at $\Omega = 0$. As shown, the resulting first-order result using Method 1 is now nearly exact at all resonance offets. In the present application, adjusting the convergence control parameter provides accuracy equivalent to one or two additional higher orders of approximation. Remarkably, this same value of $c_0$ works well for a rectangular 180° pulse (not shown) as well as 90° EBURP-2, 90° Q5, and 180° REBURP and WURST inversion pulses (vide infra).

In contrast to the results of Method 1, the power series for $y(t)$ obtained using Method 2 with $c_0 = -1$, even to third-order in $\Omega$, is accurate for $\beta(\tau_p)$ only to slightly more than $\Omega/\omega_1 = 1$. When $c_0 = -0.925$, the third-order power series has improved accuracy for resonance offsets up to nearly $\Omega/\omega_1 = 2$. However, further increases in accuracy at larger resonance offsets require very large orders of approximation $N$ in Eq. (42). For example, extending the accuracy of the power series for $\beta(\tau_p)$ to offsets $\Omega/\omega_1 = 3.5$ requires $N = 50$. The differences between the results of Method 1 and Method 2 reflects the inevitable shortcomings of power series and perturbation approaches when the expansion parameter is not small.

A more challenging example is given by the 90° EBURP-2 pulse (Geen and Freeman, 1991). In principle, the integrals in Eqs. (26) and (29) can be performed analytically, because the pulse shape is expressed as a Fourier series (as are other pulses in the BURP (Geen and Freeman, 1991) and SNOB (Kupče et al., 1995) families). In practice, the number of terms that must be calculated becomes very large and numerical integration is much more efficient. Calculations using Method 1 are shown in Fig. 2. With $c_0 = -1$, the fifth-order approximation is extremely accurate compared with numerical calculations using Eqs. (1-4) (Fig. 2a-c). With $c_0 = -0.925$ (Fig. 2d-f), even the small deviations observed for the fifth-order HAM approximation are eliminated and the third-order result is accurate except at the edge of the excitation band. In contrast, perturbation theory or power-series expansions (Method 2) are extremely poor at reproducing $\beta(\tau_p)$, essentially failing as soon as $\Omega$ is non-zero (not shown). The accuracy of the Method 1 approximations over the full range of resonance offets shows that HAM, with appropriate choice of linear operator and starting functions, can provide approximate solutions valid far outside the range of perturbation theory.

The Gaussian Q5 90° pulse (Emsley and Bodenhausen, 1992) has a more complicated amplitude modulation profile than the EBURP-2 pulse and requires higher orders of approximation to obtain accurate results. Results obtained for Method 1 with fifth- and seventh-order approximations are shown in Fig. 3. The seventh-order results is highly accurate for both $c_0 = -1$ and $c_0 = -0.925$. The choice of $c_0 = -0.925$ has a remarkable effect in increasing the accuracy the fifth-order approximation to nearly that of the seventh-order result.

The application of HAM is not limited to 90° pulses nor to amplitude-modulated pulses. Figure 4 shows the performance of Method 1 for the 180° REBURP (Geen and Freeman, 1991) and WURST-20 inversion (Kupče and Freeman, 1995) pulses. As for the EBURP-2 pulse, the fifth-order approximation for the REBURP pulse is highly accurate for both $c_0 = -1$ and $c_0 = -0.925$. The third-order approximation also is highly accurate when $c_0 = -0.925$. The WURST-20 pulse uses a linear frequency shift, generated by applying a quadratic phase shift during the pulse, and is an example of a phase-modulated or complex waveform. Again, the more complicated waveform requires higher order approximation, but eleventh-order, with $c_0 = -1$, or ninth-order, with $c_0 = -0.925$, results are highly accurate.

Method 2 yields a power series for $y(t)$. If $c_0 = -1$, the resulting series is identical to the power series expansion obtained from perturbation theory (Li et al., 2014), while $c_0 = -0.925$ provides additional accuracy. However, as noted above, the power series requires very high orders $N$ to obtain accuracy comparable to results from modest orders using Method 1. Thus, Method 1 is much more powerful for general calculations; however, the power series leads to a convenient expression for the near-resonance phase shift $\gamma(\tau_p)$. The first-order power series for $y(t)$, assuming $c_0 = -1$, yields:

$$
\begin{aligned}
f(t) &= \frac{\sin\left[\frac{\delta(t)}{2}\right] + i\Omega \int_0^t \cos\left[\frac{\delta(t)}{2} - \frac{\delta(t')}{2}\right] \sin\left[\frac{\delta(t')}{2}\right] dt'}{\cos\left[\frac{\delta(t)}{2}\right] - i\Omega \int_0^t \sin\left[\frac{\delta(t')}{2} - \frac{\delta(t')}{2}\right] \sin\left[\frac{\delta(t')}{2}\right] dt'} \\
&\approx \tan\left[\frac{\delta(t)}{2}\right] \left(1 + i\frac{\Omega}{\sin[\delta(t)]} \int_0^t \sin[\delta(t')] dt'\right)
\end{aligned}
\tag{47}
$$

in which the second equality is the expansion to first order in $\Omega$ and the resulting trigonometric functions have been simplified. This result is identical to the previously reported result from first-order perturbation theory (Li et al., 2014). The argument of

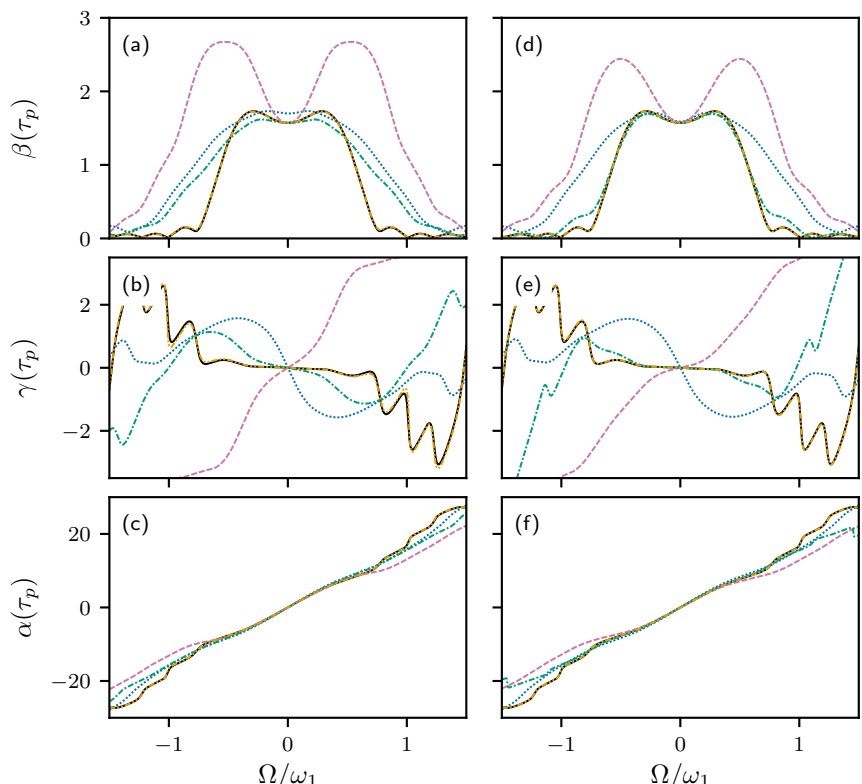

**Figure 2.** HAM approximations for 90° EBURP-2 pulse.(black) Numerical calculation of Euler angles $\alpha(\tau_p)$, $\beta(\tau_p)$, and $\gamma(\tau_p)$ using Eqs. (1-4). (blue, dotted) First-order, (reddish-purple, dashed) second-order, (green, dash-dotted) third-order, and (orange, dash-dot-dotted) fifth-order HAM results using Method 1. Results are shown for (a,b,c) $c_0 = -1$ and (d,e,f) $c_0 = -0.925$. The numerical calculation and fifth-order HAM curves are nearly indistinguishable.

the first-order approximation of $f(t)$ is a good estimate of the phase $\gamma(\tau_p)$ of the transverse magnetization following the pulse. As noted above, the phase $\alpha(t)$ is obtained by repeating the calculation with the time-reversed pulse. Therefore, as concluded from pertubation theory, an amplitude-modulated shaped pulse acts as an ideal rotation of angle $\beta(\tau_p)$ preceded and followed by time delays $\tau_\alpha$ and $\tau_\gamma$ over the frequency range for which the first-order approximation holds (Lescop et al., 2010; Li et al., 2014):

$$\tau_\alpha = \frac{1}{\sin\left[\delta(\tau_p)\right]} \int_0^{\tau_p} \sin\left[\delta(\tau_p - t')\right] dt' \tag{48}$$

$$\tau_\gamma = \frac{1}{\sin\left[\delta(\tau_p)\right]} \int_0^{\tau_p} \sin\left[\delta(t')\right] dt' \tag{49}$$

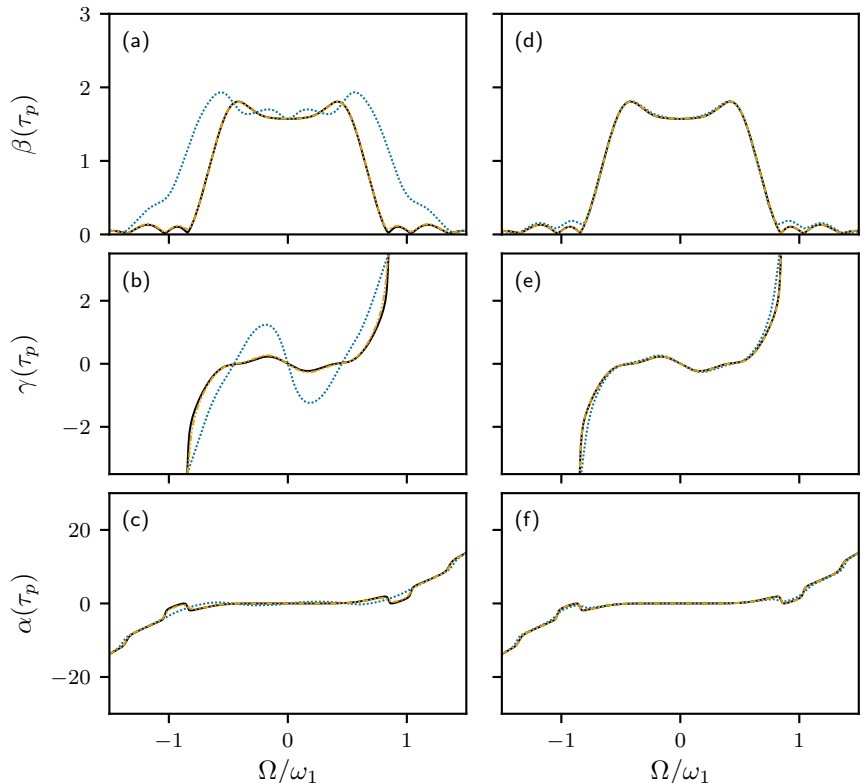

**Figure 3.** HAM approximations for 90° Q5 pulse. (black) Numerical calculation of Euler angles $\alpha(\tau_p)$, $\beta(\tau_p)$, and $\gamma(\tau_p)$ using Eqs. (1-4). (blue, dotted) fifth-order and (orange, dash-dot-dotted) seventh-order HAM results using Eqs. (26) and (29). Results are shown for (a,b,c) $c_0 = -1$ and (d,e,f) $c_0 = -0.925$. The numerical calculation and seventh-order HAM curves are nearly indistinguishable.

For a 90° pulse, the above equations can be written compactly as:

$$\tau_\alpha + i\tau_\gamma = \int_0^{\tau_p} e^{i\delta(t')} dt' \tag{50}$$

The ratios $\tau_\alpha/\tau_p$ and $\tau_\gamma/\tau_p$ are the average projections of a unit vector onto the $z$-axis and $-y$-axis respectively over the duration of the pulse (for a vector is oriented along the $z$-axis at time 0).

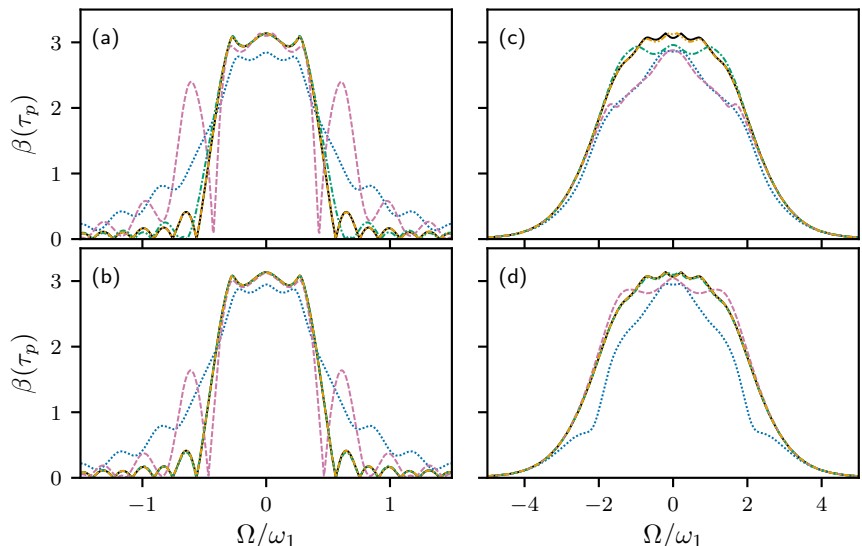

**Figure 4.** HAM approximations for (a,b) REBURP and (c,d) WURST-20 inversion pulses. (black) Numerical calculation of Euler angle $\beta(\tau_p)$ using Eqs. (1-4). (a,b) (blue, dotted) First-order, (reddish-purple, dashed) second-order, (green, dash-dotted) third-order, and (orange, dash-dot-dotted) fifth-order HAM results using Method 1. (c,d) (blue, dotted) fifth-order, (reddish-purple, dashed) seventh-order, (green, dash-dotted) ninth-order, and (orange, dash-dot-dotted) eleventh-order HAM results using Method 1. Results are shown for (a,c) $c_0 = -1$ and (b,d) $c_0 = -0.925$. (a,b) The numerical calculation and (a,b) fifth-order and (c,d) eleventh-order HAM curves are nearly indistinguishable.

The above explications have focused on solutions to the transformed Riccati equation, Eq. (8). However, HAM also could be applied directly to the original Riccati equation Eq. (5). For example, by analogy to the above appproaches, choosing

$$\mathcal{N}[g(t)] = \frac{dg(t)}{dt} - \frac{1}{2}\omega^+(t)g^2(t) - i\Omega g(t) - \frac{1}{2}\omega^-(t) \tag{51}$$

$$\mathcal{L}[g(t)] = \frac{dg(t)}{dt} - i\Omega g(t) \tag{52}$$

$$f_0(t) = \tan\left[\frac{\delta(t)}{2}\right] \tag{53}$$

in which $f_0(t)$ is the exact solution for $\Omega = 0$, yields a series solution:

$$f(t) = \tan\left[\frac{\delta(t)}{2}\right] + \sum_{n=1}^{N} f_n(t) \tag{54}$$

The first-order result is obtained from the first-order deformation equation:

$$\frac{df_1(t)}{dt} - i\Omega f_1(t) = -ic_0\Omega f_0(t) \tag{}$$

$$f_1(t) = -ic_0\Omega e^{i\Omega t}\int_0^t e^{-i\Omega t}\tan\left[\frac{\delta(t')}{2}\right]dt' \tag{55}$$

However, additional terms in the series lack the simple iterative structure shown in Eqs. (29) and (41), because of the increasing complexity of the higher-derivatives of $\Phi^2(t; q)$ that must be calculated for the $n$th order deformation equation. For example, the differential equations for the next two terms in the series for $f(t)$ become:

$$240 \quad \frac{df_2(t)}{dt} - i\Omega f_2(t) = c_0\{\frac{df_1(t)}{dt} - i\Omega f_1(t) - \omega(t)f_0(t)f_1(t)\} \tag{56}$$

$$\frac{df_3(t)}{dt} - i\Omega f_3(t) = c_0\{\frac{df_2(t)}{dt} - i\Omega f_2(t) - 2\omega(t)f_0(t)f_2(t) - \omega(t)f_1^2(t)\} \tag{57}$$

In addition, results obtained using Eq. (30) to obtain $f(t)$ from $y(t)$ generally are more accurate than results obtained by direct calculation of $f(t)$, at the same order of approximation. Thus, in this particular application, use of HAM with the transformed Riccati equation, Eq. (8), yields more convenient expressions. Nonetheless, this example demonstrates the particular power

of HAM in directly converting the solution of a non-linear differential equation into a series of linear first-order differential equations, which always can be solved by integration (Liao, 2012).

For many applications, the Euler angles for a shaped pulse are easily obtained from Eqs. (1-4). However, calculations using Eqs. (26) (29), and (30) (Method 1) are extremely efficient. In Python 3.6, the seventh-order HAM approximation for the Q5 pulse is approximately 20-fold faster than direct calculation using Eqs. (1-4). Thus, these approximations may be particularly

useful for computational design of radiofrequency pulses, in which many interations of a search or optimization routine are necessary (Gershenzon et al., 2008; Li et al., 2011; Nimbalkar et al., 2013; Asami et al., 2018).

The Euler angle representation is particularly convenient because, once calculated, the Euler angles can be used to determine the outcome of a shaped pulse applied to arbitrary initial magnetization. The Ricatti equation can be extended to incorporate radiation damping, but not relaxation, as discussed by Rourke (Rourke, 2002). However, the Euler angles can serve to generate

the initial approximations for a second application of HAM to obtain approximate solutions to the Bloch equations for particular initial conditions, including relaxation. In the following, $\Omega(t) = \Omega$ is assumed to be fixed and only amplitude-modulated pulses $\omega(t)$ with $x$-phase are considered (these assumptions can be relaxed as needed). The Bloch equations for a pulse applied with $x$-phase can be written in the form:

$$\frac{d}{dt}\hat{\mathbf{M}}(t) = -\mathbf{\Gamma}(t)\hat{\mathbf{M}}_z(t) + \begin{bmatrix} 0 \\ 0 \\ e^{R_2 t}R_1 M_0 \end{bmatrix} \tag{58}$$

in which,

$$\hat{\mathbf{M}}(t) = \begin{bmatrix} \hat{M}_x(t) \\ \hat{M}_y(t) \\ \hat{M}_z(t) \end{bmatrix} \tag{59}$$

$$\mathbf{\Gamma}(t) = \begin{bmatrix} 0 & \Omega & 0 \\ -\Omega & 0 & \omega_x(t) \\ 0 & -\omega_x(t) & -(R_2 - R_1) \end{bmatrix} \tag{60}$$

$M_k(t) = e^{-R_2 t} \hat{M}_k(t)$ are the Cartesian components of the magnetization, and $M_0$ is the equilibrium magnetization. The use of transformed variables $\hat{\mathbf{M}}(t)$ rather than $\mathbf{M}(t)$ simplifies the following discussion. The linear operator is chosen as

$\mathcal{L}[\mathbf{g}(t)] = d\mathbf{g}(t)/dt$, in which $\mathbf{g}(t) = [g_x(t), g_y(t), g_z(t)]^T$ is an arbitrary matrix function. The non-linear operator is

$$N[\mathbf{g}(t)] = \frac{d\mathbf{g}(t)}{dt} + \mathbf{\Gamma}(t)\mathbf{g}(t) - \begin{bmatrix} 0 \\ 0 \\ e^{R_2 t} R_1 M_0 \end{bmatrix} \tag{61}$$

The initial zeroth-order approximations for HAM are $\hat{\mathbf{M}}_0(t) = \mathbf{M}_0(t)$ and are given by the solutions to the Bloch equations in the absence of exchange for initial equilibrium magnetization $[0, 0, M_0]^T$. The initial approximations are calculated by using the Euler angles determined from Method 1 described above. Thus,

$$\frac{d}{dt}\hat{\mathbf{M}}_0(t) + \Gamma(t)\hat{\mathbf{M}}_0 = 0 \tag{62}$$

and the system of first-order deformation equations yield:

$$\frac{d}{dt}\hat{\mathbf{M}}_1(t) = -c_0 \begin{bmatrix} 0 \\ 0 \\ (R_2 - R_1)\hat{M}_{z0}(t) + e^{R_2 t} R_1 M_0 \end{bmatrix} \tag{63}$$

giving $\hat{M}_{x1}(t) = \hat{M}_{y1}(t) = 0$, and:

$$\hat{M}_{z1}(t) = -c_0 \left[ (R_2 - R_1) \int_0^t \hat{M}_{z0}(t')dt' + \frac{R_1}{R_2}(e^{R_2 t} - 1)M_0 \right]$$

$$= -c_0 \left[ (R_2 - R_1)t < \hat{M}_{z0}(t) > + \frac{R_1}{R_2}(e^{R_2 t} - 1)M_0 \right] \tag{64}$$

The first-order approximation of magnetization during the pulse is given by:

$$\mathbf{M}(t) = e^{-R_2 t} \begin{bmatrix} M_{x0}(t) \\ M_{y0}(t) \\ M_{z0}(t) + \hat{M}_{z1}(t) \end{bmatrix} \tag{65}$$

At this level of approximation, relaxation of transverse magnetization depends simply on $R_2$, while relaxation of $M_z(t)$ depends on the average $z$-magnetization during the pulse (calculated in the absence of relaxation). For macromolecules,

$R_2 >> R_1$ typically and the term proportional to $R_1/R_2$ is small.

The $n$th-order deformation equation leads to the following expression for $n > 1$:

$$\hat{\mathbf{M}}_n(t) = (1 + c_0)\hat{\mathbf{M}}_{(n-1)}(t) - c_0 \int_0^t \mathbf{\Gamma}(t')\hat{\mathbf{M}}_{(n-1)}(t')dt' \tag{66}$$

If $c_0 = -1$, the above recursive expressions can be written compactly as:

$$\hat{\mathbf{M}}_n(t) = (-1)^{n-1} \int\limits_0^t \boldsymbol{\Gamma}(t_{n-1})dt_{n-1} \int\limits_0^{t_{n-1}} \boldsymbol{\Gamma}(t_{n-2})dt_{n-2}... \int\limits_0^{t_2} \boldsymbol{\Gamma}(t_1)\hat{\mathbf{M}}_1(t_1)dt_1 \tag{67}$$

For a rectangular pulse applied to equilibrium magnetization (with magnitude set to unity for convenience), the initial approximations are:

$$\mathbf{M}_0(t) = \begin{bmatrix} [1 - \cos(\omega_e t)]\cos\theta\sin\theta \\ -\sin(\omega_e t)\sin\theta \\ \cos(\omega_e t)\sin^2\theta + \cos^2\theta \end{bmatrix} \tag{68}$$

and

$$\hat{M}_{z1}(t) = -c_0(R_2 - R_1)\left[\frac{1}{\omega_e}\sin(\omega_e t)\sin^2\theta + t\cos^2\theta\right] - c_0\frac{R_1}{R_2}(e^{R_2 t} - 1) \tag{69}$$

In this case, $\boldsymbol{\Gamma}(t) = \boldsymbol{\Gamma}$ and the series of approximations given in Eq. (67) can be summed to give:

$$\mathbf{M}(t) = e^{-R_2 t}\left\{\mathbf{M}_0(t) + \int\limits_0^t e^{\boldsymbol{\Gamma}(t-t')}\begin{bmatrix} 0 \\ 0 \\ (R_2 - R_1)M_{z0}(t') + e^{R_2 t'}R_1 M_0 \end{bmatrix}dt'\right\} \tag{70}$$

and yields identical results as direct integration of the Bloch equations. Equations (65), (67), and (70) explicitly show the effect of relaxation as a perturbation of the evolution of magnetization in the absence of relaxation.

Figure 5 shows the magnetization components for rectangular $90°$, $180°$, $270°$, and $360°$ nominal on-resonance pulses in the absence and presence of relaxation. Calculations were perfomed in the absence of relaxation using Eq. (68), and in the presence of relaxation using the HAM approximations, Eqs. (69) and (70). The first-order HAM approximation is surprisingly accurate for moderate values of $R_2$, except for cases in which $< \hat{M}_{z0}(t) >= 0$, such as the on-resonance $360°$ pulse. The above expressions display the fundamental dependence of relaxation during a pulse applied to equilbrium magnetization on the time-average $z$-magnetization.

## 2.3 Conclusion

Fast, accurate methods for solving differential equations have widespread application in NMR spectroscopy. The present work has illustrated the Homotopy Analysis Method (Liao, 2012) for approximating solutions for differential equations by application to the Riccati differential equation for the Euler angle representation of radiofrequency pulse shapes and to solutions of the Bloch equations incorporating relaxation. The freedom to select the linear operator, lowest-order approximate solution, convergence control parameter, and auxiliary function is powerful in obtaining series solutions that are highly accurate for low orders of approximation and efficient to calculate or that provide qualitatively convenient series, allowing physical insight. It can be expected that Homotopy Analysis Method will find other applications in NMR spectroscopy.

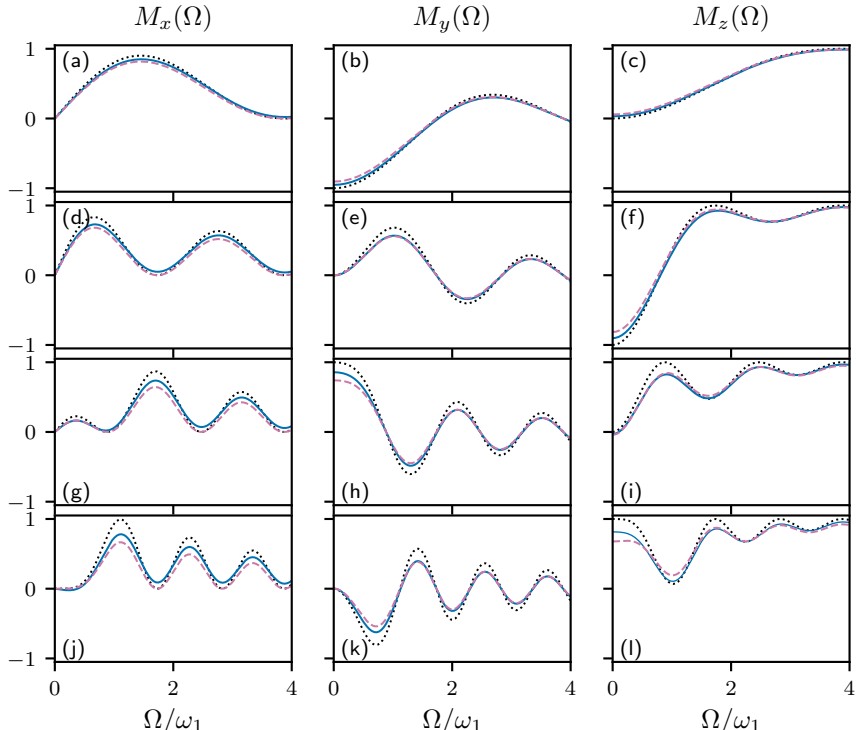

**Figure 5.** HAM approximations for rectangular (a,b,c) $90°$, (d,e,f) $180°$, (g,h,i) $270°$, and (j,k,l) $360°$ pulses applied to initial $z$-magnetization. Values of (a,d,g,j) $M_x(\Omega)$, (b,e,h,k), $M_y(\Omega)$, and (c,f,i,l) $M_z(\Omega)$ are shown as functions of resonance offset $\Omega$. (black, dotted) Magnetization components in absence of exchange using Eq. (68), (reddish-purple, dashed) first-order HAM approximation of the Bloch equations using Eq. (69), and (blue, solid) exact HAM solution of the Bloch equations using Eq. (70). Calculations used $\omega_1/(2\pi) = 250$ Hz, $R_1 = 2\,\mathrm{s}^{-1}$, $R_2 = 100\,\mathrm{s}^{-1}$, and $c_0 = -1$.

*Code and data availability.* RMarkdown and bibtex files are provided as supplementary material. The RMarkdown file contains Python code for all calculations described in the paper.

*Author contributions.* A.G.P. conceived the project. Theoretical derivations, numerical calculations and writing of the paper were performed by T.C. and A.G.P.

*Competing interests.* The authors declare that they have no competing interests.

*Acknowledgements.* This work was supported by National Institutes of Health grant R35 GM130398 (A. G. P.). Some of the work presented here was conducted at the Center on Macromolecular Dynamics by NMR Spectroscopy located at the New York Structural Biology Center, supported by a grant from the NIH National Institute of General Medical Sciences (P41 GM118302). A.G.P. is a member of the New York Structural Biology Center. This paper is dedicated to Prof. Robert Kaptein on the occasion of his 80th birthday.

315

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
