# Peer review of "Approximate Representations of Shaped Pulses Using the Homotopy Analysis Method"

_Magnetic Resonance, 2021_

## Author Comment (AC1)

1. The matrix form of the propagator given in Equation 1 is expressed in which basis?

   The matrix from of the propagator is expressed in the Cartesian basis. This has been clarified in the text.

2. I may have missed it but the author should mention that all calculations are performed in the rotating frame.

   The reviewer is correct and this point has been clarified in the text.

3. Are omega_x and omega_y, first introduced below Equation 4, the radiofrequency field amplitudes along the x and y axes of the rotating frame?

   The reviewer is correct and this has been clarified in the text.

4. It is only just below Equations 13 and 14 that the authors introduce the idea that the value of the homotopy defined in Equation 10 should be zero. For easier understanding by readers, I believe it could be mentioned with Equations 10-12 that we aim at solving $H = 0$, right?

   The desired solution to the differential equation is obtained if homotopy defined in Eq. 12 is equal to 0 when $q = 0$ and when $q = 1$. The homotopy need not be zero for other values of $q$. Thus, we believe that the order of presentation in the paper is appropriate in making clear that that homotopy is not required to be zero for all values of $q$.

5. In equation 13, should the last remaining q be replaced by 0?

   The reviewer is correct and this has been fixed in the text.

6. The derivation of Equations 24 and 27 does not seem trivial. Can the authors mention at least the method used to obtain these two expressions?

   A term proportional to $y_i(t)$ is absent from Eq 24. Thus, the homogeneous solution can be found by two integrations and the inhomogeneous solution found by variation of parameters. We have added a reference to a standard text on differential equations. The same is true for Eq. 27.

7. The function delta(t) is introduced discreetly in Equation 33. However, it is used extensively and could benefit from being defined in a separate equation.

   The reviewer probably refers to Eq. 25, not 33. We have made the suggested change.

8. Just below Equation 38, the fact that y(t) is defined by a power series in Omega, which can be large, is intriguing and raises the question: does the series converge? The results section seems to confirm the doubt about quick convergence of the series. Maybe this aspect could be discussed in more detail either below Equation 38 or in the results section.

   One of the main advantages of the HAM method is that approximations can be obtained that converge much faster than traditional power series, depending on choice of the linear operator and $y_0(t)$. Of course, for particular choices, one might obtain a power series as shown in Eq. 38.

The discussion of the results for a square pulse illustrates the slow convergence of the power series for the rotation angle: the $50^{th}$ order power series approximation performs worse than the second order Method 1. The performance for more complicated pulse shapes is usually even worse, as noted in the text.

In contrast to the poor accuracy for the rotation angle, the power series provides a very useful expression for the linear phase evolution during a shaped pulse, as originally presented by Li and coworkers.

9. It can be hard to read the figures as many overlapping curves are "shown". The authors may wish to make an effort to make as many curves visible as possible, for instance by using dashed or dotted lines for the curves sitting on top of another one.

We thank the reviewer for this helpful suggestion.

10. The alpha (tp) label seems to have drifted too far to the left in Figures 2 and 3.

We have redrawn the figures.

11. The authors conclude their abstract and conclusion with the mention that the HAM can be applied to many problems in NMR. Without saying too much, could the authors give a few hints, beyond the optimization of shaped pulses? Could the effects of relaxation be introduced by modifying the Ricatti equation, for instance?

The Ricatti equation can be modified to include radiation damping, but not relaxation (Rourke, D.E. (2002), Solutions and linearization of the nonlinear dynamics of radiation damping. Concepts Magn. Reson., 14: 112-129. https://doi.org/10.1002/cmr.10005). HAM however is generally applicable to solution of single or systems of differential equations, provided suitable linear operators, initial approximations $y_0(t)$, and auxiliary functions $H(t)$ can be found.

12. A couple of comments on references: (i) in the introduction, the reference to Cavanagh et al. 2007 for shaped pulses is very relevant, but the authors may wish to add a few other references to reviews or original work on this topic; (ii) it seems that SNOB pulses are mentioned but the original reference is not cited.

We thank the reviewer for noticing this oversight.

13. "the" is repeated in line 121.

We thank the reviewer for the careful reading of the paper.

---

## Author Comment (AC2)

1. The basis sets for the matrix representations should be given.

   The matrix from of the propagator is expressed in the Cartesian basis. This has been clarified in the text.

2. Some of the equations, like Eq. (24), are not obvious, pointers at derivation may be given.

   A term proportional to $y_i(t)$ is absent from Eq 24. Thus, the homogeneous solution can be found by two integrations and the inhomogeneous solution found by variation of parameters. We have added this information and a reference to a standard text on differential equations. The same is true for Eq. 27.

3. It is written that HAM works well with an appropriate choice of a linear operator and stating functions. Are these given in the manuscript for the examples illustrated? How easy is to fix these for an arbitrary pulse? What are the guiding parameters in this regard?

   The choices of liner operator and starting approximation are given in the text for each application of HAM. For Method 1, the linear operator is given in Eq. 19 and the initial starting function given in the following text. In the present case we chose, mostly by trial-and-error, the linear operator and starting function so that the differential equation for $y_1(t)$ had a very simple form and the equations for $y_n(t)$ were "special" second-order differential equations with known analytical solutions. The monograph by Liao discusses choices of these parameters in a number of applications. Certainly, the choice of these parameters is the difference between success and failure.

4. Any comments on the quaternions and Euler angles calculation for the pulse methods illustrated here?

   We are not sure what specifically is being asked by the reviewer. The quaternion and Euler angle methods are highly accurate in calculating the performance of shaped pulses and the purpose of the paper is not to criticize these techniques. Indeed, the 'exact' calculations provided in the paper for comparison to approximations using HAM were calculated using the Euler angle approach.

5. When the authors say that the HAM-Riccati approach may work well for other cases in NMR, what do they have in mind?

   The Ricatti equation can be modified to include radiation damping, but not relaxation (Rourke, D.E. (2002), Solutions and linearization of the nonlinear dynamics of radiation damping. Concepts Magn. Reson., 14: 112-129. https://doi.org/10.1002/cmr.10005). HAM however is generally applicable to solution of single or systems of differential equations, provided suitable linear operators, initial approximations $y_0(t)$, and auxiliary functions $H(t)$ can be found.

6. In the current manuscript, although elegant examples are given, I am not sure how the approach can be used to make the schemes better. Perhaps this can be explained in the text.

> As noted in the text, the efficiency of the Method 1 calculations may be useful in optimization of new pulse shapes, in which many iterative steps must be taken.

7. On a semantic level, I am not sure what is actually meant by theoretical magnetic resonance. It is essentially an experimental field with solid inputs from theory. We may not want to just keep calculating some parameters from sophisticated equations which may have no practical relevance.

> We have removed the word "theoretical" from the Introduction. As noted above, the HAM approach may be useful for optimization of new pulse shapes. However, in some cases, the HAM results (as in other analytic approximations) may provide new insights. For example, Method 1 gives $f(t)$ essentially as the ratio of sums of iterated integrals of w(t). The iterated integrals are related to the Fourier integrals used by Warren in a perturbation expansion using average Hamiltonian theory. This correspondence suggests a connection between the two approaches that might prove useful, although beyond the scope of the present paper. Finally, one purpose of the paper is to introduce to NMR spectroscopists a powerful approach for approximating the solutions to systems of differential equations. We anticipate that other spectroscopists will find interesting applications of the approach in their own work.